# Objectively measured physical activity levels and adherence to physical activity guidelines in people with multimorbidity—A systematic review and meta-analysis

**Lars Bo Jørgensen**[1,2,3]*, **Alessio Bricca**[1,3], **Anna Bernhardt**[1], **Carsten B. Juhl**[3], **Lars Hermann Tang**[1,4], **Sofie Rath Mortensen**[1,5], **Jonas Ahler Eriksen**[1], **Sisse Walløe**[1,6], **Søren T. Skou**[1,3]

**1** Department of Physiotherapy and Occupational Therapy, The Research Unit PROgrez, Næstved-Slagelse-Ringsted Hospitals, Slagelse, Region Zealand, Denmark, **2** Department of Physiotherapy and Occupational Therapy, Zealand University Hospital, Roskilde, Region Zealand, Denmark, **3** Department of Sports Science and Clinical Biomechanics, Research Unit for Musculoskeletal Function and Physiotherapy, University of Southern Denmark, Odense, Denmark, **4** The Department of Regional Health Research, University of Southern Denmark, Odense, Denmark, **5** Department of Sports Science and Clinical Biomechanics, Research Unit for Exercise Epidemiology, Centre of Research in Childhood Health, University of Southern Denmark, Odense, Denmark, **6** Department of Clinical Research, Research Unit OPEN, University of Southern Denmark, Odense, Denmark

* labjo@regionsjaelland.dk

## Abstract

### Objective

To determine levels of objectively measured physical activity (PA) and the proportion of adults with multimorbidity that adheres to PA guidelines.

### Methods

All studies, where PA was measured at baseline using an activity monitor in an adult (≥18 years) multimorbid (≥80% of the population had ≥2 chronic conditions) population. A systematic literature search was performed in Medline, EMBASE, CINAHL, CENTRAL, ClinicalTrials.gov, opengrey.eu and google.com from inception up until 18th of January 2022. Risk of bias was assessed with a modified version of the Quality Assessment Tool for Quantitative Studies. A random-effects meta-analyses was performed to estimate daily minutes of sedentary behavior (SB), light PA (LPA), moderate PA (MPA), moderate to vigorous PA (MVPA) and steps. Proportions adhering to PA guidelines was narratively synthesized. Certainty of evidence was determined using The Grading of Recommendations Assessment, Development and Evaluation (GRADE) approach.

### Results

Fifteen studies (2,172 participants) were included. The most frequent combination of conditions were type 2 diabetes and hypertension (six studies). Participants spent a daily average of 500.5 (95% CI: 407.1 to 593.9) minutes in SB, 325.6 (95% CI: 246.4 to 404.7 minutes in

**Data Availability Statement:** All relevant data are within the paper and its Supporting information files.

**Funding:** LBJ is funded by two grants from The Næstved-Slagelse-Ringsted Hospitals Research Fund and Region Zealand (Program grant, Exercise First). LBJ and LT are both funded by a grant from The Danish Health Confederation through the Development and Research Fund (grant No 2703). The Næstved-Slagelse-Ringsted Hospitals Research Fund (webpage only exists in Danish) https://www.regionsjaelland.dk/sundhed/geo/slagelsesygehus/Om-sygehuset/Forskning/Sider/Forskningsfonde.aspx Region Zealand (Exercise First) https://www.regionsjaelland.dk/sundhed/geo/slagelsesygehus/Afdelinger/Reumatologisk-afdeling/forskning-progrez/Sider/Exercise-First-projektet.aspx The Danish Health Confederation through the Development and Research Fund https://sundhedskartellet.dk/english/ STS is currently funded by a program grant from Region Zealand (Program grant, Exercise First) and two grants from the European Union's Horizon 2020 research and innovation program, one from the European Research Council (MOBILIZE, grant agreement No 801790) and the other under grant agreement No 945377 (ESCAPE). AB is funded by the European Research Council under the European Union's Horizon 2020 research and innovation programme (MOBILIZE, grant agreement No 801790). European Union's Horizon 2020 research and innovation program https://ec.europa.eu/info/research-and-innovation/funding/funding-opportunities/funding-programmes-and-open-calls/horizon-europe_en European Research Council https://ec.europa.eu/info/research-and-innovation/funding/funding-opportunities/funding-programmes-and-open-calls/horizon-europe/european-research-council_en The funders had no role in study design, data collection and analysis, decision to publish, or preparation of the manuscript.

**Competing interests:** The authors have declared that no competing interests exist.

LPA and 32.7 (95% CI: 20.2 to 45.3) minutes in MVPA. The mean daily number of steps was 5,145 (95% CI: 4264 to 6026) for people in free-living conditions. The proportion adhering to PA guidelines ranged widely (7.4% to 43%). All studies were rated as at high risk of bias and the certainty of evidence was very low.

## Conclusions

PA levels and adherence varied from low to above guideline recommended levels for adults with chronic conditions, depending on PA intensity. The very low certainty of evidence calls for high quality studies focusing on detailed descriptions of PA behavior in people with multimorbidity.

## PROSPERO registration number

CRD42020172456.

## 1. Introduction

The prevalence of people having two or more chronic conditions—commonly referred to as 'multimorbidity'—is estimated to be 33% in the general population [1, 2] and more than half of all adults with a chronic condition are multimorbid [3]. Multimorbidity is not only a burden for the individual, but also for society with an almost exponential association between the number of chronic conditions and associated health care costs [4]. Multimorbidity is considered to be the next global health priority [5], due to an expected increase in prevalence of multimorbidity in the near future [6], impacting the lives of millions of people worldwide [7].

Physical inactivity represents an important target in the growing burden of multimorbidity as it increases the risk of poorer health, development of further chronic conditions [8] and death [9]. It is well known that physical activity (PA) and exercise are effective in preventing and treating chronic conditions such as type 2 diabetes, heart and pulmonary diseases and osteoarthritis [8, 10]. Furthermore, PA and exercise appear both safe and beneficial in improving physical and psychosocial health in people with multimorbidity [11, 12].

The World Health Organization (WHO) recommends that adults with chronic conditions should perform at least 150–300 minutes of moderate-intensity aerobic PA, 75 to 150 minutes of vigorous-intensity PA (or an equivalent combination) per week [13]. However, despite the well-documented benefits of PA, many people do not adhere to PA guidelines [14]. This proportion increases with the number of chronic conditions i.e. the level of PA is lower among people with multimorbidity [15–17]. A recent study found that nearly 68% of older adults with multimorbidity fail to meet the level of PA recommended by the WHO [18].

Unfortunately, and in spite of emerging evidence there is still limited knowledge on PA level in people with multimorbidity. Furthermore, no overview of the level of objectively measured PA and adherence to PA guidelines is available, although objectively measured PA is considered more accurate and less susceptible to bias than self-reported PA [19–22]. Therefore, this systematic review and meta-analysis aimed to determine 1) the level of objectively measured PA in people with multimorbidity in free-living conditions and 2) the proportion that adheres to the WHO PA guidelines for people with chronic conditions.

## 2. Methods

This review followed the recommendations for performing systematic reviews as described by Cochrane [23]. Reporting of the review was done in accordance with the Preferred Reporting Items for Systematic Reviews and Meta-analyses 2020 (PRISMA 2020) guidelines [24] (S1 File).

### 2.1 Eligibility criteria

**2.1.1 Study design and participants.**   Studies where PA was measured objectively (e.g., activity monitor such as a pedometer or accelerometer) in an adult multimorbid population (mean age ≥18 years) were included. Only baseline PA data was extracted. The population was considered multimorbid if ≥80% had two or more of the following chronic conditions; osteoarthritis (knee or hip), chronic heart disease (heart failure or ischemic heart disease), hypertension (systolic blood pressure ≥140 and diastolic blood pressure ≥90 and/or intake of anti-hypertensive medications), type 2 diabetes mellitus (T2DM), chronic obstructive pulmonary disease (COPD), depression, anxiety, or chronic low back pain (LBP). If multimorbidity was reported as a cumulative numerical score or an index score, they were excluded unless the number of conditions exceeded 80% of the conditions that we used to define multimorbidity in this review. These conditions were chosen as they are among the leading causes of global disability, affect hundreds of millions of people around the world, and often co-exist [25]. Studies were furthermore excluded if PA was measured for a period of less than 7 days or had reported less than 2 valid days of PA measurement.

### 2.2 Information sources

Literature was searched in the scientific databases Medline and EMBASE (via Ovid), CINAHL (via EBSCOhost) and CENTRAL, from inception up until 18[th] of January 2022. Grey literature was searched using a grey literature checklist developed by Godin and colleagues [26] on ClinicalTrials.gov and via the webpages opengrey.eu and google.com. Furthermore, reference lists of included studies were hand searched for eligible studies and citation tracking performed on included studies in Web of Science (WoS).

### 2.3 Search strategy

The search strategy was developed for Medline and then customized for the remaining databases (S2–S5 Files). Search strategies were developed individually for clinical.trials.gov, opengrey.eu and google.com (S6 File). No limits were set on language.

### 2.4 Selection process

Records were transferred to Covidence software [27] and duplicates removed using the Covidence software. Two reviewers (LBJ and AnB) independently screened records for eligibility on title and abstract and resolved any conflicts by discussion. For full text screening, the review team was expanded to include four additional members (AB, SM, SW and JE), forming three teams consisting of two reviewers. The remaining studies were full text screened for eligibility individually as follows (SW/JE: 60 studies, AB/SM: 60 studies and LBJ/AnB: 240 studies. Conflicts were resolved by discussion within the review team or by involvement of a more experienced senior review team-member (STS or CJ) if needed.

## 2.5 Data collection process

Data from included studies were extracted independently by two reviewers (LBJ and AB) using a data extraction form developed in Excel by LBJ. The extraction procedure was pilot tested on three studies and data compared among the reviewers in order to implement any adjustments to the data extraction form. No adjustments were made. Data were then extracted from the remaining studies and compared. Consensus on extracted data was reached through discussion. If multiple papers were published based on data from the same study, the paper with the largest sample size was used and other papers excluded.

## 2.6 Data items

Data was extracted regarding study characteristics, participant and outcome data.

Study characteristics included author, year, country of origin of the study, and study design (e.g., observational). Participant data included population number (n =), mean age, gender (percentage of female), body mass index (BMI), ethnicity and number of conditions, severity, and diagnosis.

Outcome data included type (e.g., pedometer or accelerometer), brand and placement of activity monitor, duration of the PA measurement (e.g., 7 days), number of valid days of measurement and definition of intensity level (e.g., cut point for moderate-vigorous PA (MVPA)). For continuous outcome (physical activity) data, mean and standard deviation, standard error, or 95% Confidence Interval was extracted of the following variables: minutes spent as sedentary behavior (SB), light PA (LPA), moderate PA (MPA), moderate to vigorous (MVPA), vigorous PA (VPA), daily steps or 'other' (i.e., activity counts, calories).

When PA was reported as weekly levels it was converted into daily levels by dividing it with the number of valid days of measurement reported in the individual study. In PA data presented with medians, these were considered equivalent to the mean, and interquartile ranges used to calculate standard deviations (SD) as recommended by Cochrane [23]. SDs of daily PA levels were converted to standard errors (SE) to perform the meta-analyses. If the exact number of participants that contributed with PA data was not reported in a study, the total number of the population was used in data conversions. In studies reporting two measures of PA (e.g., bouted and unbouted PA), the number of participants were split in two groups of smaller sample sizes, and results reported as two separate estimates as recommended by Cochrane [23].

The following items, although reported in the PROSPERO registration, were not presented, as they were not consistently reported in the retrieved studies: socioeconomic status, physical function, mental health, quality of life, wear time (hours per day), epoch length and cut point for non-wear time.

## 2.7 Study risk of bias assessment

Risk of bias was assessed using a quality assessment tool designed specifically for this systematic review (S7 File). The tool was inspired by the Quality Assessment Tool for Quantitative Studies, developed for use in the Effective Public Healthcare Panacea Project (EPHPP) (https://www.ephpp.ca/quality-assessment-tool-for-quantitative-studies/). Questions not considered relevant for the aims of this review were deleted. The deleted questions were: Rating of study design, Confounders, Withdrawals and drop-outs, Intervention integrity and Analyses. Furthermore, options to answer, and wording was altered to fit the aims of the review e.g., *'Are the individuals selected to participate in the study likely to be representative of the target population?'* was replaced with *'How representative was the study participants of the multimorbid target population?'* This is a common procedure and has been done in several previous studies [28]. Risk of bias was assessed through six questions divided into three sections; selection bias,

blinding (performance bias) and data collection methods (measurement bias). Each section was rated as strong, moderate, or weak (blinding could only be rated strong or weak) based on ratings of the questions in the section. Each study was given a global rating based on the ratings of the three sections. A low-quality rating (one or more weak ratings) was considered high risk of bias, moderate-quality rating (only strong or moderate ratings) as moderate risk of bias and high-quality rating (only strong ratings) as low risk of bias. Risk of bias assessment was performed by two review teams consisting of four reviewers LBJ/SM and SW/JE. Each team member assessed studies independently and compared ratings within their review team. To assist the rating, a dictionary was incorporated after each section of the quality assessment tool explaining the rationale behind rating. This was inspired by the original assessment tool. In case of disagreements consensus was reached through discussion.

## 2.8 Effect measures

**2.8.1 Estimate measures.**   Average minutes of daily activity spent as SB, LPA, MPA, MVPA, VPA and steps was assessed in separate meta-analyses.

## 2.9 Synthesis of results

**2.9.1 Physical activity.**   Meta-analyses were performed for minutes of activity spent as SB, LPA, MPA, MVPA, VPA and daily steps in free-living conditions. A random-effects model was used given the heterogeneous population included. Results were presented in forest plots with overall estimates of the PA level including subgroup-analyses of bouted and unbouted PA. Subgroup-analyses were performed in order to investigate the possible differences in PA arising from analyzing data as bouted versus unbouted, which have been suggested in earlier studies [29–31]. Statistical heterogeneity was assessed using $I^2$ statistics and interpreted according to the Cochrane recommendations as a continuous measure where an I-squared value of 0% indicates no inconsistency, and an I-squared value of 100% indicates maximal inconsistency. All statistical analyses were performed using Stata 17 (StataCorp. 2021. *Stata Statistical Software*: *Release 17*. College Station, TX: StataCorp LLC) using the 'meta' command.

If studies did not report enough data to be included in meta-analysis and authors did not provide the necessary data, a narrative synthesis of the results was performed in accordance with the Cochrane recommendations [23].

**2.9.2 Adherence to physical activity guidelines.**   The proportion of people adhering to PA guidelines was summarized in a narrative synthesis as there were only three studies with heterogeneous populations reporting proportions, or insufficient data available to calculate proportions. Therefore, meta-analysis was deemed not appropriate in line with the Cochrane recommendations for performing meta-analysis [23].

**2.9.3 Reporting bias assessment.**   Authors of potentially eligible studies were contacted about outcome and population characteristics. For example, asking for measure of variance (e.g. standard deviation) regarding PA measures and prevalence of chronic conditions in the reported population when it was only reported that the population had multimorbidity. All contact to author was done by e-mail to ensure the greatest response rate possible [32]. Initially an e-mail was sent to the corresponding author. If an answer was not received within fifteen workdays, the last author was contacted, and the deadline extended with fifteen more days. If no reply was received, the study was excluded.

**2.9.4 Sensitivity analysis.**   Sensitivity analysis was performed by removing studies presenting median values to check the robustness of the findings, give that mean and median values were pooled in the same meta-analyses as recommended by Cochrane [23].

**2.9.5 Certainty assessment.** The certainty (overall quality) of evidence was determined for each meta-analyses using the GRADE (Grading of Recommendations Assessment, Development and Evaluation) approach [33] for prognostic studies, given the nature of the included studies and in line with Iorio et al. [34]. Five domains; risk of bias, inconsistency, indirectness, imprecision, and publication bias were assessed individually by two reviewers (LBJ and AB). In case of disagreements, consensus was reached through discussion.

# 3. Results

## 3.1 Study selection and characteristics

A total of 17,537 records were identified through databases and registers. Another 19 records were identified by searching the webpages Opengrey.eu and Google.com. Eight additional records were identified by screening references of published systematic reviews focusing on multimorbidity.

Of 360 full text screened reports, 345 were excluded (S8 File) with the main exclusion criteria being non-multimorbid populations or non-peer reviewed reports. No further studies were identified through hand search of reference lists of included studies or WoS citation tracking.

Nine studies [35–43] appeared eligible after full text screening but had not reported PA data in a format making it possible to extract for meta-analyses (i.e. missing SD or SE), or lacked data on the proportion of population that was multimorbid. Authors of these studies were contacted. Four authors replied providing additional information, leading to inclusion of three additional studies [36, 42, 43] and exclusion of one study due to the population not being multimorbid [35]. The flow of studies is presented in Fig 1.

Fifteen studies were included [36, 42–55] with a total of 2,172 participants. Publication year ranged from 2011 to 2021. The most common study designs were randomized controlled trials [42, 45, 47, 48, 50, 51, 53] and observational studies [43, 44, 46, 49, 52, 53, 55]. The most frequent combination of conditions was type 2 diabetes (T2DM) and hypertension (six studies) [36, 43, 49, 50, 52, 55]. No studies had populations where more than 80% of the participants had more than two chronic conditions.

Accelerometers were the most commonly used measurement method (11 of 15 studies) [36, 42–44, 46–48, 51–54], and steps the most frequent way to report the PA level (8 of 15 studies) [42, 45, 49, 50, 52–55].

Three studies reported adherence (or data making it possible to calculate such) to physical activity guidelines [36, 52, 54]. Study characteristics of included studies are presented in Tables 1 and 2.

## 3.2 Risk of bias in studies

One third of the included studies were rated as 'weak' in all three parts of the risk of bias assessment, mainly due to studies failing to describe the validity and reliability of the measurement method used. Overall, all included studies were of low quality (high risk of bias) (see Table 3).

## 3.3 Results of syntheses

**3.3.1 Physical activity level.** No studies reported minutes of PA spent at all intensity levels and daily steps. The pooled mean daily minutes spent in SB and LPA was 500.5 (95% CI: 407.1 to 593.9) and 325.6 (95% CI: 246.4 to 404.7) (Figs 2 and 3). The pooled mean daily minutes of MVPA was 32.7 (95% CI: 20.2 to 45.3) (Fig 4). MPA and VPA were only reported in one study [54] and meta-analysis of these activity levels therefore not performed.

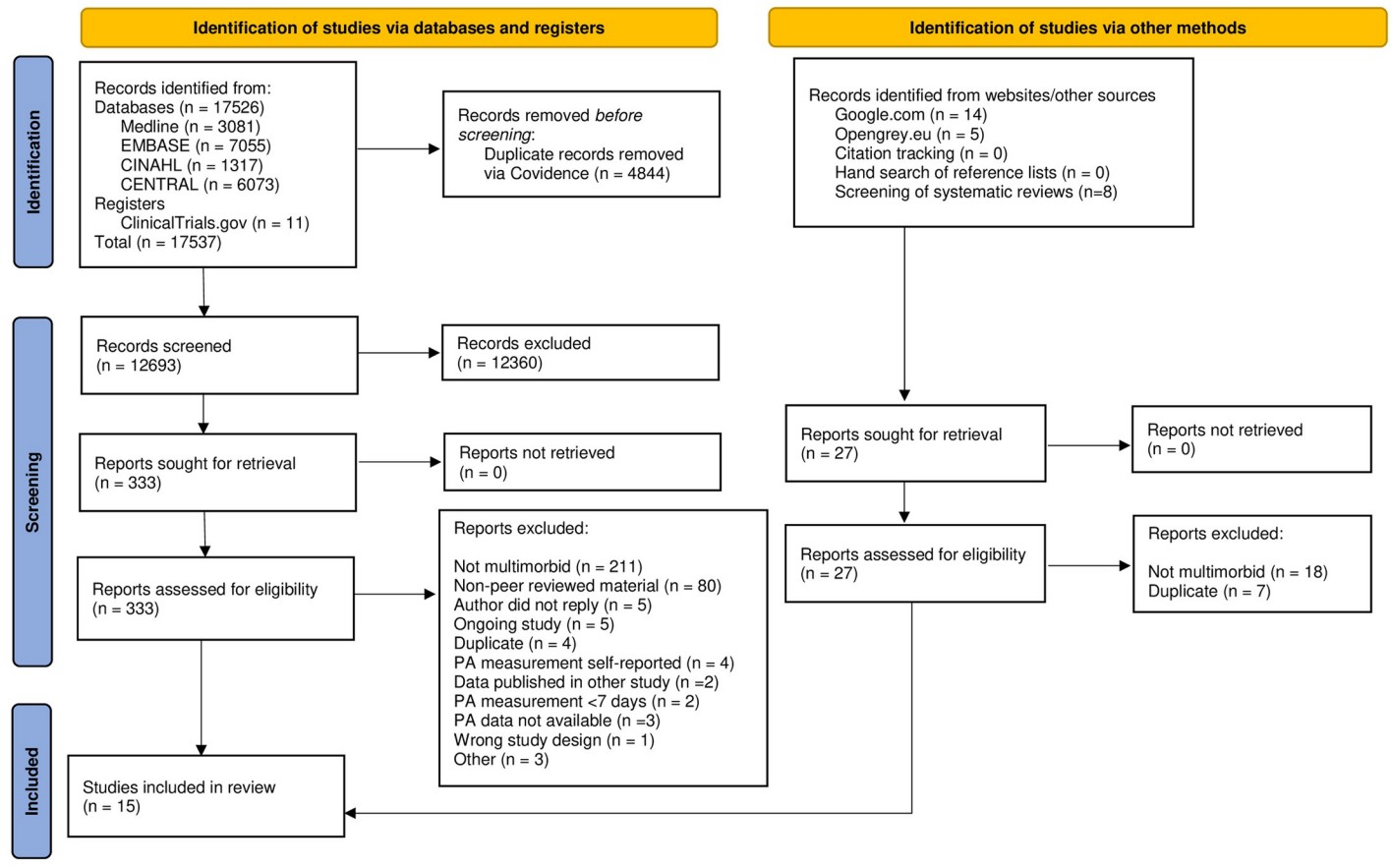

**Fig 1. Flow of studies.**

The pooled mean of daily steps was 5145 (95% CI: 4264 to 6026) (Fig 5). One study, not included in meta-analysis, included hospitalized participants reporting a median daily number of steps of 1170 (IQR: 523 to 2580).

Two studies [46, 51] reported PA as other than minutes or steps (activity counts and active hours) (Table 2). All meta-analyses showed high heterogeneity and subgroups analyses on PA stratified as bouted or unbouted did not explain the inconsistency of the results for all the meta-analyses (Figs 2–4). The sensitivity analysis removing studies reporting median values displayed similar results as the main analyses (S1–S3 Figs).

**3.3.2 Adherence to physical activity guidelines.** The proportion that adhered to physical activity guidelines was reported in three studies and ranged from 7.4% to 43%. The populations had combinations of T2DM and hypertension (two studies) [36, 52] and depression and anxiety (one study) [54].

## 3.4 Certainty of evidence

Certainty of evidence was assessed as being very low for all meta-analyses (Table 4).

## 4. Discussion

Results from this systematic review and meta-analysis showed that the daily level of MVPA exceeded what is recommended by the WHO for people with chronic conditions, and that the

**Table 1. Conditions and selected demographic variables in the included studies.**

| First author, year, country, study design | (n =) | Age | BMI | Gender (% ♀) | Ethnicity (%) | Condition (%) | Condition definition | Severity |
|---|---|---|---|---|---|---|---|---|
| **Piette, 2011, USA, RCT** | | | | | | | | |
| Usual care group | 146 | 56 | 38 | 50 | White (84) Black (9) Other (7) | T2DM and depression (100) | T2DM: identified via electronic records/ self-reported. Depression: PHQ-9 depression score ≥11 | T2DM: NR Depression BDI>29: 34% |
| Usual care + CBT group | 145 | 55.1 | 37.3 | 51 | | | | T2DM: NR Depression BDI>29: 32% |
| **Izawa, 2013, Japan, Observational** | 95 | 56.8 | 22.9 | 11.4 | NR | Heart failure and depression (100) | Heart failure: NYHA classification. Depression: SF-36, MH <68 points | NYHA I: 33% NYHA II: 46% NYHA II: 21% LVEF<40% Depression: NR |
| **Paula, 2014, Brazil, RCT** | | | | | | | | |
| Control group | 20 | 62.5 | 30.2 | 70 | White (90) | T2DM and hypertension (100) | T2DM: outpatients from hospital department. Hypertension: office BP ≥140/90 mm Hg and daytime ABPM ≥135/85 mm Hg | (duration years) T2DM: mean 16.1 Hypertension: mean 18.4 |
| Intervention group | 20 | 61.8 | 28.6 | 40 | White (80) | | | T2DM: mean 16.9 Hypertension: mean 16.9 |
| **Freedland, 2015, USA, RCT** | | | | | | | | |
| Usual care group | 79 | 55.5 | 32.6 | 33 | White (72.2) | Heart failure and depression (100) | Heart failure: NYHA classification. Depression: current major depressive episode and BDI-II score ≥14 | LVEF<45%: 59% NYHA I-II: 57% NYHA III: 43% BDI-II: mean 29.6 |
| Usual care + CBT group | 79 | 56.2 | 34.7 | 40 | White (54.4) | | | LVEF<45%: 48.7% NYHA I-II: 58.2% NYHA III: 41.8% BDI-II: mean 30.7 |
| **Schneider, 2016, USA, Pilot RCT** | | | | | | | | |
| Exercise group | 15 | 53.3 | 34.5 | 100 | White (86.7) Black/African American (6.7) Other: (6.7) | T2DM and depression (100) | T2DM: Inadequately controlled T2DM. Depression: doctor diagnosed major depressive disorder as defined by the SCID-IV criteria | T2DM: NR; BDI-II: mean 18.5; HRSD: mean 15.7 |
| Enhanced usual care group | 14 | 53.6 | 34.7 | 100 | White (85.7) American Indian/ Alaskan native (14.3) | T2DM and depression (100) | T2DM: Inadequately controlled T2DM. Depression: doctor diagnosed major depressive disorder as defined by the SCID-IV criteria | T2DM: NR; BDI-II: mean 21.6 HRSD: mean 17.4 |
| **Zucatti, 2017, Brazil, Observational** | 151 | 61.1 | 29,8 | 64 | White (77 | T2DM (100) and hypertension (92) | T2DM: NR. Hypertension: mean of office BP measurement >140/90 on two occasions or use of antihypertensive medication | (duration years) T2DM: mean 14.3 Hypertension. NR |
| **Lambert 2018, UK, Pilot RCT** | 62 | 38.1 | NR | 84 | NR | Depression and anxiety (100) | Depression: PHQ-8. Anxiety: GAD-7 (no cut off scores used) | (duration years) PHQ-8: mean 14.6 GAD-7: mean 11.8 |
| **Moreira 2018, Brazil, Observational** | | | | | | | | |
| Vitamin D deficient group | 66 | 65 | 30 | 53 | White (85) | T2DM and hypertension (100) | T2DM: hospital diagnosed history of T2DM. Hypertension: office BP≥140/90 mm Hg and/or current use of anti-hypertensive medication | (duration years) T2DM: median 12 Hypertension: median 15 |
| Vitamin D non-deficient group | 50 | 65 | 31 | 62 | White (82) | | | T2DM: median 11 Hypertension: median 14 |
| **Hult 2019, Sweden, Observational** | 210 | 70 | 29.2 | 34 | Caucasian (100) | T2DM (100) and hypertension (82) | T2DM: Self-reported. Hypertension: use of anti-hypertensive medication | T2DM: NR Hypertension: NR |

(*Continued*)

**Table 1.** (Continued)

| First author, year, country, study design | (n =) | Age | BMI | Gender (% ♀) | Ethnicity (%) | Condition (%) | Condition definition | Severity |
|---|---|---|---|---|---|---|---|---|
| **Oliveira 2019, Portugal, Observational** | | | | | | | | |
| <10m/sec group | 45 | 55,8 | 27.1 | 10.6 | NR | Ischemic heart disease (100) and hypertension (94.1) | Ischemic heart disease: patients recruited from hospital department. Hypertension: average of three BP measurements | Ischemic heart disease: NR Hypertension: NR |
| >10m/sec group | 23 | | | | | | | |
| **Reddy 2020, USA, Observational** | | | | | | | | |
| QOL worst group | 133 | 66 | 37 | 48 | NR | Heart failure (100) and hypertension (89) | Heart failure: doctor diagnosed and objective evidence**. Hypertension: NR | LVEF ≥50%: 100% NYHA class II-IV Hypertension: NR |
| QOL intermediate group | 134 | 71 | 33 | 58 | | Heart failure (100) and hypertension (82) | | |
| QOL best group: | 141 | 70 | 32 | 50 | | Heart failure (100) and hypertension (84) | | |
| **Whipple 2020, USA, Clinical trial** | 19 | 72.1 | 30.5 | 36.8 | White (89.5) | T2DM (100) and hypertension (94.7) | T2DM: NR<br><br>Hypertension: NR | T2DM: NR Hypertension: NR |
| **Schlenk 2021, USA, RCT** | 182 | 64.7 | 34 | 73.1 | White (73.1) | Osteoarthritis, knee–OA and hypertension (100) | OA: Clinical examination. Hypertension: Intake of antihypertensive medication | OA: WOMAC pain subscale: mean 5.3 Hypertension: NR |
| **Correia, 2021, Brazil, Observational** | 121 | 68 | 27.8 | NR | NR | T2DM (100) and hypertension (92) | T2DM: medical history and use of medication. Hypertension: average of the last two of three measurements at the arm with highest pressure | NR |
| **Holber 2021, USA, RCT** | 222 | 64 | NR | 43 | White (49) | Heart failure and depression (100) | Heart failure: inpatients recruited from hospital departments. Depression: PHQ-9 depression score ≥10 | LVEF≤45%: 100% |
| | | | | | | | | NYHA class |
| | | | | | | | | II: 28% |
| | | | | | Non-white (51) | | | III: 60% |
| | | | | | | | | IV: 12% |
| | | | | | | | | PHQ-9: median 13 |

Abbreviations:

RCT: Randomized controlled trial, CBT: cognitive-behavioral therapy, NR: not reported, PHQ: Patient Health Questionnaire, BDI: Beck Depression Inventory, SF-36, MH: 36-item short form health survey, mental health items, NYHA: New York Heart Association, BP: Blood Pressure, ABPM: daytime ambulatory blood pressure, LVEF: Left Ventricular Ejection Fraction, T2DM: Type 2 Diabetes Mellitus, OA: osteoarthritis, HRSD: Hamilton Rating Scale for Depression, EX: Exercise, EUC: Enhanced Usual Care, SCID: Structured Clinical Interview for DSM (statistical manual for mental disorders) disorders, GAD: General Anxiety Disorder scale, WOMAC: Western Ontario and McMaster Universities Arthritis Index.

*population consisted of cardiac patients with heart failure, myocardial infarction, coronary artery bypass grafting or valve replacement,

**Invasive hemodynamics, elevated natriuretic peptide levels or echocardiographic diastolic dysfunction together with chronic use of diuretic medication.

few studies specifically reporting proportions that adhere to PA guidelines found wide variation in adherence (7,4–43%). Results also showed that participants spent on average more than eight hours per day in SB and had a number of daily steps equivalent to a low active lifestyle among adults [13, 56]. The results should be interpreted with caution since few studies evaluating objectively measured PA levels in multimorbid populations were available, and due to the high risk of bias of the included studies and very low overall certainty of evidence.

Although challenged by the lack of available evidence, higher levels of daily MVPA and LPA and lower levels of SB were found than previously reported in people with multimorbidity. In a Canadian cohort study, Hains-Monfette et al. [57] reported that people with two

**Table 2. Physical activity measurements in the included studies.**

| Author, year | Activity monitor (duration) | Brand | Placement | Valid day definition | Valid days (number) | PA definition | Daily physical activity level Mean (SD)/[SE] |
|---|---|---|---|---|---|---|---|
| **Piette 2011** | Pedometer (7 days) | Omron Hj-720 ITC | NR | NR | NR | Steps | Control group: 3139 (2361) |
| | | | | | | | Intervention group: 3226 (1860) |
| **Izawa 2013** | Uniaxial accelerometer (8 days) | Kenz lifecorder | Waist (above either leg) | NR | NR | Steps | 5020.1 (2735.92) |
| **Paula 2014** | Pedometer (7 days) | Yamax Digi-Walker CW200 | NR | NR | 3 | Steps | Control group: 5848 (2827) |
| | | | | | | | Intervention group: 6294 (2544) |
| **Freedland 2015** | Accelerometer (7 days) | Respironic Actiwatch AW-16 | Wrist (non-dominant hand) | NR | NR | Other (7 day average activity counts) | 92.7 (55.1) |
| **Schneider 2016** | Accelerometer (7 days) | Actigraph 7164 WAM | Hip | ≥10 hours | 3 | Minutes[1] | Control group: MVPA*: 13.2 (12.1) |
| | | | | | | | Intervention group: MVPA*: 12.6 (12.7) |
| **Zucatti, 2017** | Pedometer (7 days) | Yamax Digi-Walker SW700 | Waist | NR | NR | Steps | 6391 (3357) |
| | | | | | | Other (km/week) | 3.1 |
| **Lambert 2018** | Triaxial accelerometer (7 days) | GENEActiv | Wrist (non-dominant hand) | ≥10 hours | 4 | Minutes[2] | LPA*: 174.3 (56) |
| | | | | | | | MPA*: 53.5 (30.2) |
| | | | | | | | VPA*: 2.9 (3.8) |
| | | | | | | | MVPA*: 8.95 [5.0] |
| | | | | | | | MVPA**: 55.2 (36.8) |
| **Moreira 2018** | Pedometer (7 days) | Yamax Digi-Walker CW200 | Waist | NR | NR | Steps | Vitamin D deficient group: 6400 (2518) |
| | | | | | | | Vitamin D non-deficient group: 4400 (2888) |
| **Hult 2019** | Triaxial accelerometer (7 days) | Actigraph GT3X | Hip (non-dominant leg) | ≥10 hours | 4 | Steps | 5904 (3038) |
| | | | | | | Minutes[3] | MVPA*: 26.7 [4.5] |
| | | | | | | | MVPA**: 39.2 [5.4] |
| **Oliveira 2019** | Accelerometer (7 days) | ActiGraph GT1M | Hip (right) | ≥8 hours | 5 | Minutes[4] | <10m/sec group: |
| | | | | | | | SB**: 455.6 [39] |
| | | | | | | | LPA**: 384.2 [46.4] |
| | | | | | | | MVPA**: 45.8 [11.2] |
| | | | | | | | >10m/sec group: |
| | | | | | | | SB**: 460.4 [47.9] |
| | | | | | | | LPA**: 377.2 [54.9] |
| | | | | | | | MVPA**: 26.8 [16.5] |
| | | | | | | Other: total minutes of PA time (per week) | <10m/sec group: |
| | | | | | | | 2238 (637) |
| | | | | | | | >10m/sec group: |
| | | | | | | | 2055 (574) |
| **Reddy 2020** | Triaxial accelerometer (14 days) | Kinetic Activity Monitor KXUD9-2050 | Hip | ≥10 hours | NR | Other (accelerometry hours active per day) | QOL worst group: 5.83 (1.4) |
| | | | | | | | QOL intermediate group: 6.45 (2.2) |
| | | | | | | | QOL best group: 6.35 (1.6) |

(*Continued*)

**Table 2.** (Continued)

| Author, year | Activity monitor (duration) | Brand | Placement | Valid day definition | Valid days (number) | PA definition | Daily physical activity level Mean (SD)/[SE] |
|---|---|---|---|---|---|---|---|
| **Whipple 2020** | Triaxial accelerometer (14 days) | Actigraph wGT3X-BT | Wrist (non-dominant hand) | ≥10 hours | 5 | Minutes[5] | SB**: 473 (101.3) |
| | | | | | | | LPA**: 102 (23.4) |
| | | | | | | | MVPA**: 74 (44.6) |
| | | | | | | Other (minutes in sedentary bouts***) | Other: 191.6 (89.7) |
| **Schlenk 2021** | Triaxial accelerometer (7 days) | Actigraph GT3X+ | Waist | NR | NR | Minutes[6] | SB*: 425.2 (104.9) |
| | | | | | | | LPA*: 333.4 (81.7) |
| | | | | | | | MVPA*: 44.7 (31.6) |
| **Correia 2021** | Triaxial accelerometer (7 days) | Actigraph GT3X/GT3X+ | Hip (right) | ≥10 hours | 4 | Minutes[7] | SB*: 675.3 (104.2) |
| | | | | | | | LPA*: 273.5 (95.5) |
| | | | | | | | MVPA*: 11.2 (14.9) |
| **Holber 2021** | Accelerometer (7 days) | SenseWear Pro | Arm (upper) | ≥10 hours | 4 | Steps | 1170 (median)**** |

Notes:

Abbreviations: NR: not reported. SB: sedentary behavior, LPA: light physical activity, MPA: moderate physical activity VPA: vigorous physical activity, MVPA: moderate to vigorous physical activity, QOL: quality of life

*: bouted,

**: unbouted,

***: periods of ≥10 minutes with less than 99 activity counts/min,

****: hospitalized population

[1] Moderate or greater intensity: ≥1952 counts/min

[2] MVPA: ≥1952 counts/min

[3] LPA: ≤1951 counts/min, moderate: 1952–5724, hard: 5725–9498, very hard: ≥9499

[4] SB: 0–99 count/min, LPA: 100–2019 counts/min, MVPA: ≥2020 counts/min

[5] SB: 0–99 count/min, LPA: 100–1951 counts/min, moderate: ≥1952 counts/min

[6] None to very low: 0–99 counts/min, LPA: 100–2019 counts/min, MVPA: ≥2020 counts/min

[7] SB: 0–100 counts/min, LPA: 101–1.040 counts/min, MVPA: ≥1.041 counts/min

chronic conditions performed a daily average of 10 minutes of MVPA, 171 minutes of LPA and 580 minutes of SB. These results were based on an adult population with somewhat similar chronic conditions (heart disease, diabetes, and cancer and/or COPD) to the ones included in this review. Direct comparison of PA levels is, however, challenged by the general lack of data on factors that could possibly affect PA levels, such as the prevalence and severity of the included conditions which has previously demonstrated to be associated with PA [58].

The mean number of daily steps found in the meta-analysis in this study were similar to studies including healthy adult general populations from Western countries measuring steps in free-living conditions [59, 60]. Considering existing evidence suggesting that people with chronic conditions are less physically active than the general population [15], the number of daily steps were expected to be lower in populations with multimorbidity compared to the general population. Importantly, results from the present study showed that people with multimorbidity only attained a daily step level equivalent to the level of 'low PA' using earlier published cut off points for healthy adults of 5000–7499 daily steps [56]. It is likely that the contradictive findings between an adequate daily MVPA level and 'low active' daily step level could be explained by the difference in participants entering the meta-analyses. Only one study [52] reported data on both MVPA and steps and interpretation of the PA levels was

**Table 3. Risk of bias assessment of included studies.**

| Study | Selection bias | | | | Blinding* | | Data collection method** | | | Study quality |
|---|---|---|---|---|---|---|---|---|---|---|
| | Population | Participation | Physical activity data | *Rating* | Blinding | *Rating* | Validity | Reliability | *Rating* | |
| Piette et al. 2011 | 2 | 2 | 2 | Moderate | 1 | Strong | 3 | 3 | Weak | **Low** |
| Izawa et al. 2013 | 3 | 1 | 1 | Weak | 3 | Weak | 1 | 1 | Strong | **Low** |
| Paula et al. 2014 | 1 | 1 | 1 | Strong | 2 | Weak | 3 | 3 | Weak | **Low** |
| Freedland et al. 2015 | 1 | 2 | 1 | Moderate | 3 | Weak | 3 | 3 | Weak | **Low** |
| Schneider et al. 2016 | 3 | 1 | 3 | Weak | 3 | Weak | 3 | 3 | Weak | **Low** |
| Zucatti et al. 2017 | 2 | 3 | 1 | Weak | 2 | Weak | 1 | 3 | Weak | **Low** |
| Lambert et al. 2018 | 3 | 2 | 1 | Weak | 3 | Weak | 1 | 1 | Strong | **Low** |
| Moreira et al. 2018 | 1 | 1 | 1 | Strong | 3 | Weak | 3 | 3 | Weak | **Low** |
| Hult et al. 2019 | 2 | 3 | 1 | Weak | 2 | Weak | 1 | 1 | Strong | **Low** |
| Oliveira et al. 2019 | 2 | 3 | 2 | Weak | 1 | Strong | 1 | 1 | Strong | **Low** |
| Reddy et al. 2020 | 2 | 1 | 1 | Moderate | 3 | Weak | 3 | 3 | Weak | **Low** |
| Whipple et al. 2020 | 3 | 3 | 3 | Weak | 3 | Weak | 1 | 3 | Moderate | **Low** |
| Schlenk et al. 2021 | 2 | 3 | 1 | Weak | 3 | Weak | 3 | 3 | Weak | **Low** |
| Correia et al. 2021 | 3 | 3 | 3 | Weak | 3 | Weak | 3 | 3 | Weak | **Low** |
| Holber et al. 2021 | 2 | 3 | 3 | Weak | 3 | Weak | 3 | 3 | Weak | **Low** |

Notes:

*equivalent to assessment of performance bias,

** equivalent to assessment of measurement bias

therefore made on different sets of participants. The difference in PA levels could therefore possibly be explained by factors such as age, the specific chronic conditions, and the severity of their conditions [61]. Due to an insufficient number of available studies, it was not possible to perform meta-regression analysis to investigate whether these and other factors affected the results.

A previous large observational study from the UK Biobank found that almost half of people with multimorbidity can be categorized as having a low physical activity level [62]. A large variation in reported proportions of adherence to PA guidelines was observed in the narrative synthesis in this study (7,4–43%). This appears conflicting to the relatively high daily level of MVPA found in the meta-analysis. Such variation is similar to findings from previous studies of self-reported PA in people with multimorbidity [61] and has also been observed in the general healthy population [63]. This appears conflicting to the relatively high daily level of MVPA found in the meta-analysis. Wide variation in PA levels have however also been demonstrated in previous studies of self-reported PA in people with multimorbidity [61] and in the general healthy population [63]. Possible explanations could be the use of different measurement methods (self-reported vs. objectively measured PA) or that selection bias was introduced in the present study and could have affected the estimates. This is likely since the included populations displayed differences in characteristics such as age or specific chronic conditions. Last, functional status of the different populations were unknown since this data was not available or inconsistently reported. If the some of the populations included in the MVPA meta-analyses had higher functional status level, this could have resulted in higher MVPA estimates as supported by previous research [64].

The GRANADA consensus statement on analytical approaches for accelerometer-determined physical behaviors [65] gives an optional recommendation to express estimates of time

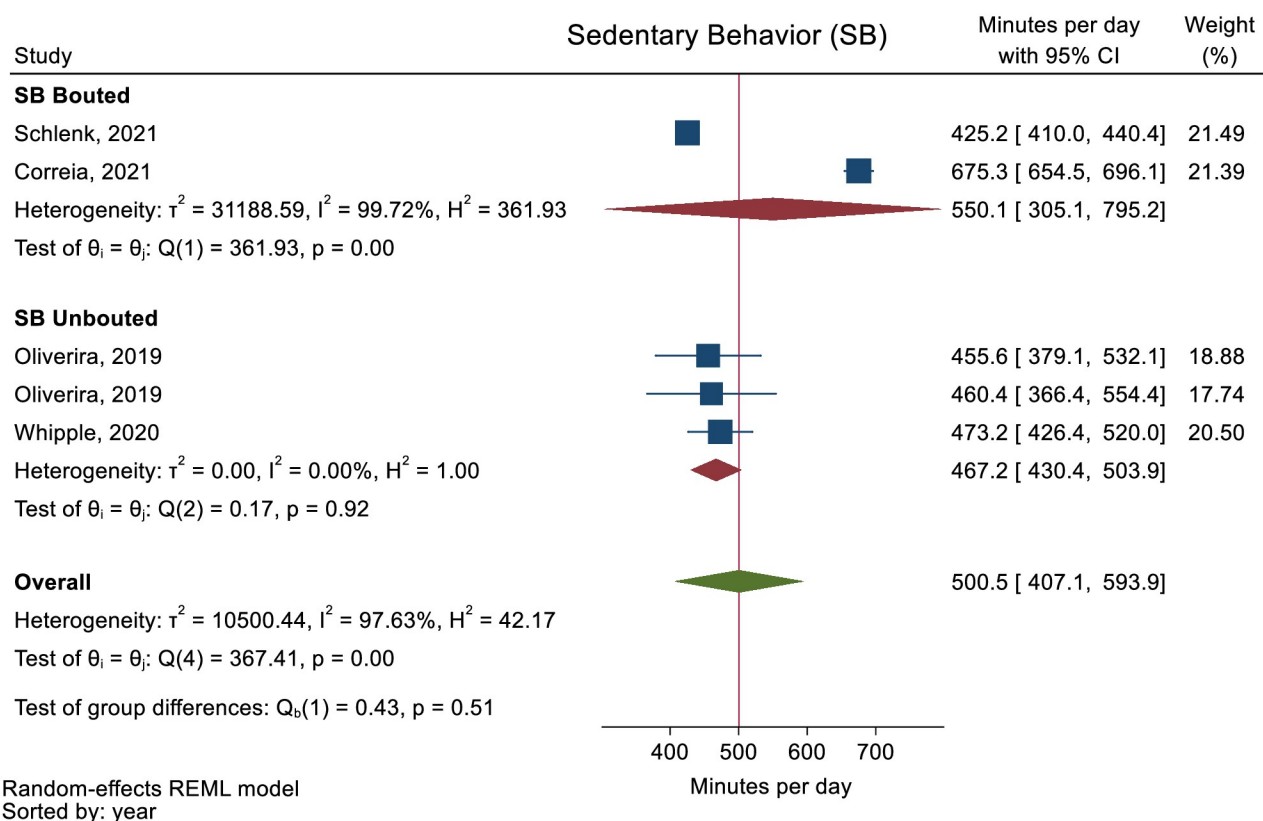

**Fig 2. Forest plot showing daily minutes of Sedentary Behavior (SB) in people with multimorbidity.**

spent in PA as bouted and unbouted. Although not statistically significant, the results from this meta-analysis showed that the number of activity minutes varied greatly between studies analyzing data on bouted versus unbouted PA. Highest levels were found in meta-analysis of unbouted data, except for SB, suggesting that unbouted data could potentially display higher activity levels. In the recent update of PA guidelines from WHO (2020) [13], recommendations that PA should be performed in bouts of ≥10 minutes were removed in recognition that all PA, regardless of length, promotes health. This is not yet fully reflected in analyses of PA data in studies being published.

## 4.1 Clinical implications and future research

The level of daily SB and steps found in this review revealed that there is still a potential to gain further health benefits by increasing PA in people with multimorbidity. These benefits could potentially be even more important in older populations, since studies have shown that older people spent the vast majority of their time in SB [66]. Furthermore, current evidence also supports that the level of PA decreases with increasing age in both healthy [67] and multimorbid populations [18]. Despite an extensive knowledge gap in the area, clinicians should therefore promote PA for people with chronic conditions and multimorbidity given the documented health benefits [10, 11]. However, they should also acknowledge the importance of primary prevention, since physical inactivity is a risk factor for development of chronic conditions and multimorbidity [8, 68]. In populations failing to adhere to PA guidelines, special attention should be given to the importance of replacing SB with PA of any intensity level as

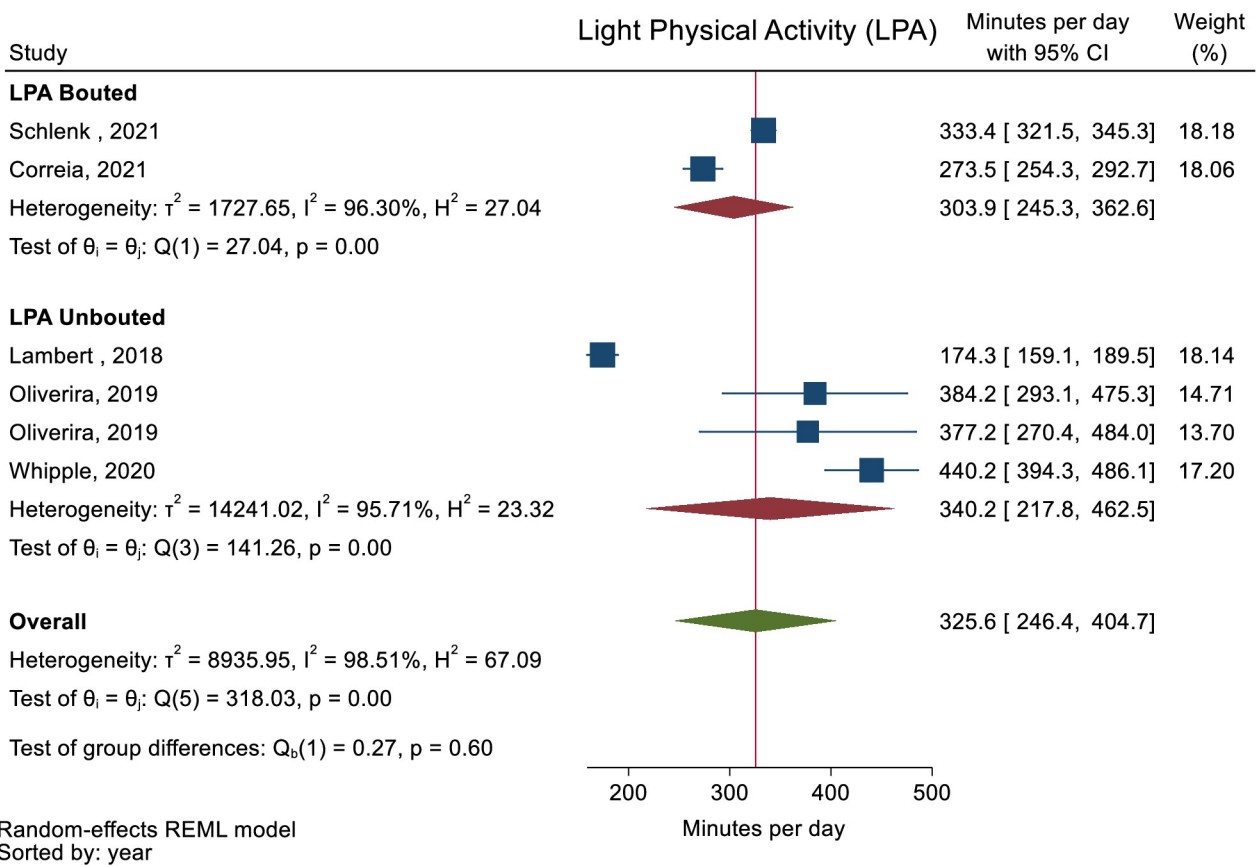

**Fig 3. Forest plot showing daily minutes of Light Physical Activity (LPA) in people with multimorbidity.**

recommended by the WHO [13]. Future studies should aim to incorporate valid measurement methods that are able to measure the full continuum of PA from SB to VPA and steps so that detailed knowledge on PA can be incorporated more when developing tailored patient-centered strategies to increase PA in people with multimorbidity.

## 4.2 Strength and limitations

In the present review, multimorbidity was defined as encompassing common, but selected, chronic conditions. This definition was inspired by earlier published literature [11] and available evidence of the benefit of PA on the included conditions [10]. A recent (2021) review found that measurements and definitions of multimorbidity is poorly reported and varies greatly (from 2 to 285 conditions) [69]. It is possible that the use of a more exhaustive list of chronic conditions would have led to more studies being included. Despite this, the authors do believe that this review managed to capture the majority of studies published by selecting conditions with a high global prevalence [25].

All meta-analyses demonstrated high statistical heterogeneity among the included studies. Furthermore, only few relevant studies were identified, all associated with a high risk of bias, leading to an overall certainty in the evidence that was very low. Additional analyses on the impact of gender, ethnicity or socioeconomic status could not be performed since too few studies reported these variables consistently. The results should therefore be interpreted with

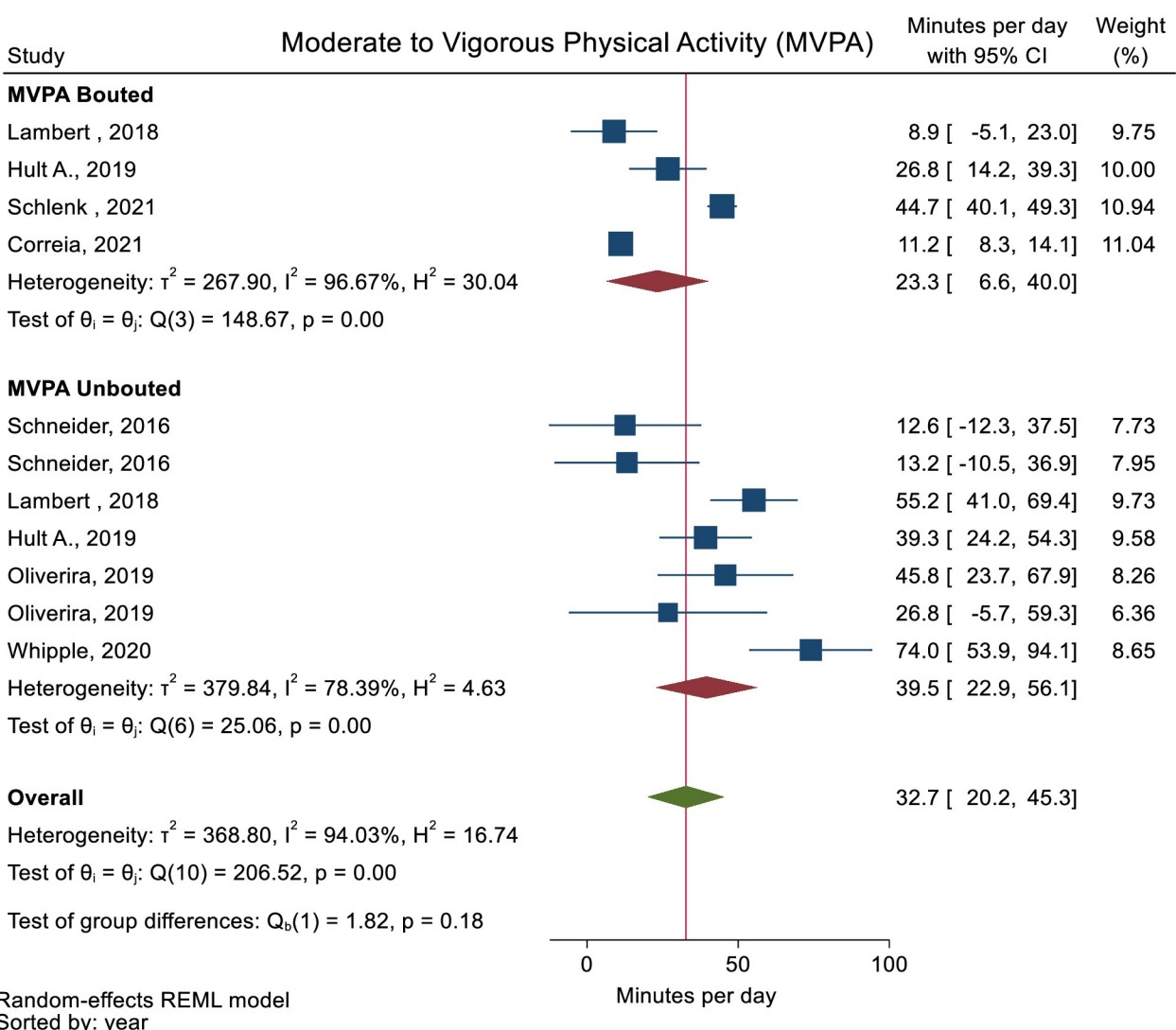

**Fig 4. Forest plot showing daily minutes of Moderate to Vigorous Physical Activity (MVPA) in people with multimorbidity.**

caution. It is highly likely that future studies will change the PA estimates identified in this review.

Studies using subjective measurement methods of PA were not included in this review, since these have demonstrated less accuracy than objective measurements [21, 22]. Subjective methods have, however, been widely used to assess PA across different populations due to low costs and accessibility. Inclusion of such studies would most likely have increased the number of included studies in the review but challenged the validity. No perfect tool exists for measuring PA, and also objective measurement methods have displayed limitations [70]. It is however suggested that researchers should incorporate appropriate objective measures specific to the PA behavior of interest when examining PA in adults in free-living conditions [19]. This recommendation is reflected in the proportion of published studies using objective measures of PA that has increased from 4% to 71% from 2006 to 2016 [71] also justifying the focus on objectively measured PA in this review.

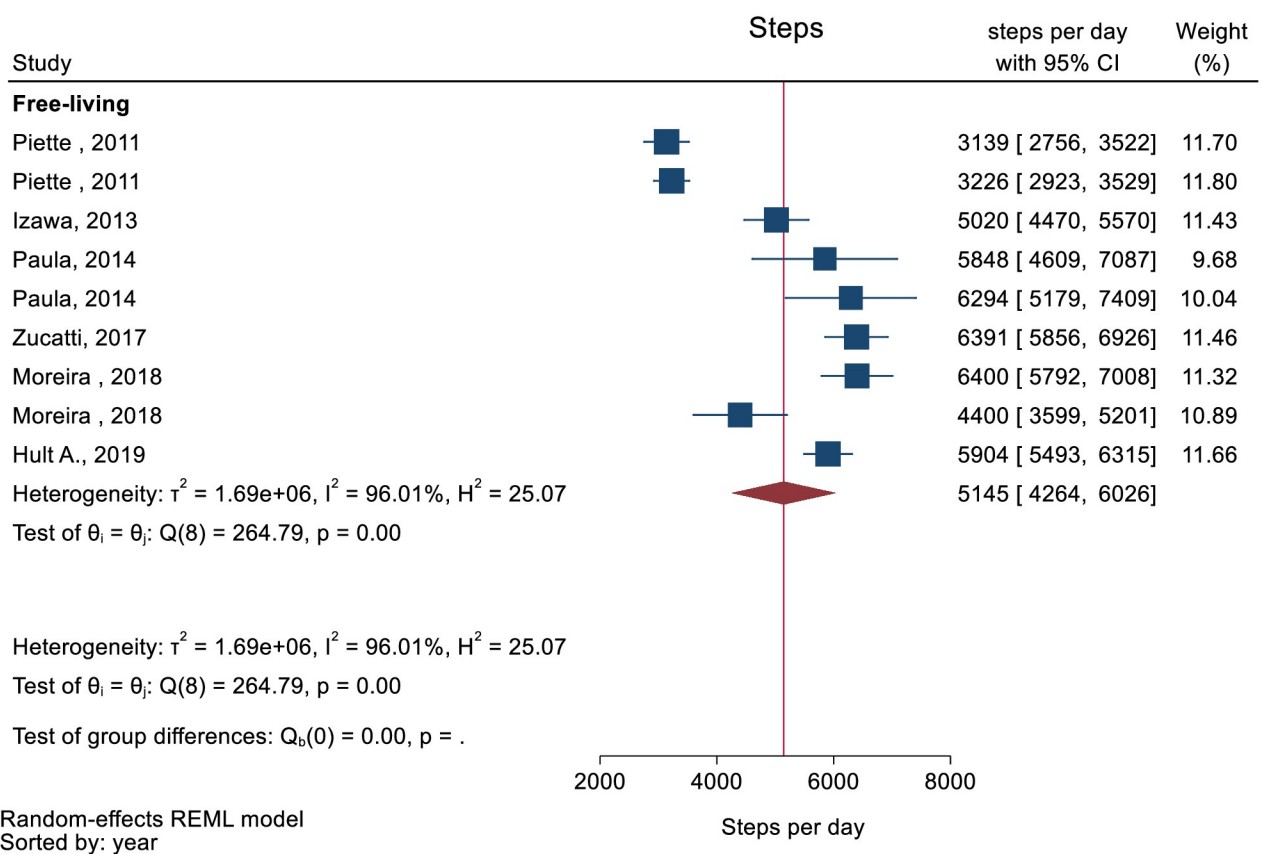

**Fig 5. Forest plot showing daily number of steps in people with multimorbidity.**

**Table 4. Certainty of evidence using the GRADE approach.**

| Number of studies | Certainty of evidence | | | | | | Certainty |
|---|---|---|---|---|---|---|---|
| | Study design | Risk of bias | Inconsistensy | Indirectness | Imprecision | Other condiserations | |
| **Steps per day** | | | | | | | |
| 6 | Observational, RCT | serious[a] | serious[b] | not serious[c] | not serious | none | ⊕◯◯◯ Very low |
| **Minutes of sedentary behavior (SB) per day** | | | | | | | |
| 4 | Observational, RCT, Clinical trial | serious[a] | serious[d,e] | serious[c] | serious[f] | none | ⊕◯◯◯ Very low |
| **Minutes of light physical activity (LPA) per day** | | | | | | | |
| 5 | Observational, RCT, Clinical trial | serious[a] | serious[e] | serious[c] | serious[f] | none | ⊕◯◯◯ Very low |
| **Minutes of moderate to vigorous physical activity (MVPA) per day** | | | | | | | |
| 7 | Observational, RCT, Clinical trial | serious[a] | serious[e] | serious[c] | serious[f] | none | ⊕◯◯◯ Very low |

Explanations:

[a]. All studies was evaluated as having high risk of bias,

[b]. Inconsistent results even after stratifying by free living and hospitalized,

[c]. Sample not representative of multimorbid populations,

[d]. 4 hour difference in sedentary time per day between the two studies included,

[e]. Inconsistent results even after stratification for bouted/unbouted,

[f]. Very wide 95% Cis.

Abbreviations: RCT = randomized controlled trial

### 4.3 Conclusion

The results of this systematic review show that the daily level of MVPA appears to exceed the level recommended in the WHO PA guidelines, while the level of daily steps is equivalent to living a low active lifestyle. Adherence to PA guidelines are currently rarely reported and varies greatly from 7,4–43%. The average time spent in SB was more than eight hours, highlighting a potential for further improvement in PA levels among people with multimorbidity, in particular less active subgroups. In general, studies investigating objectively measured PA in populations with multimorbidity were few in numbers, associated with a high risk of bias and a very low overall certainty in the evidence calling for a cautious interpretation of the results. The review highlights an urgent need for further high-quality studies providing detailed descriptions of PA behavior among people with multimorbidity.

## Supporting information

**S1 File. PRISMA checklist.**
(PDF)

**S2 File. Search strategy for Medline via OVID.**
(PDF)

**S3 File. Search strategy for EMBASE via OVID.**
(PDF)

**S4 File. Search strategy CIHNAL via EBSCOhost.**
(PDF)

**S5 File. Search strategy CENTRAL.**
(PDF)

**S6 File. Search strategy for grey literature.**
(PDF)

**S7 File. Quality assessment tool for quantitative studies.**
(PDF)

**S8 File. Studies excluded in full text screening.**
(PDF)

**S1 Fig. Sensitivity analysis with studies presenting sedentary behavior data with medians omitted.**
(TIF)

**S2 Fig. Sensitivity analysis with studies presenting light physical activity data with medians omitted.**
(TIF)

**S3 Fig. Sensitivity analysis with studies presenting moderate to vigorous physical activity data with medians omitted.**
(TIF)

**S1 Data. Data extraction form including data.**
(XLSX)

## Author Contributions

**Conceptualization:** Lars Bo Jørgensen, Anna Bernhardt, Carsten B. Juhl, Lars Hermann Tang, Søren T. Skou.

**Data curation:** Lars Bo Jørgensen, Alessio Bricca, Anna Bernhardt, Carsten B. Juhl, Sofie Rath Mortensen, Jonas Ahler Eriksen, Sisse Walløe.

**Formal analysis:** Lars Bo Jørgensen, Alessio Bricca, Carsten B. Juhl, Sofie Rath Mortensen, Jonas Ahler Eriksen, Sisse Walløe, Søren T. Skou.

**Investigation:** Lars Bo Jørgensen, Alessio Bricca, Anna Bernhardt, Søren T. Skou.

**Methodology:** Lars Bo Jørgensen, Alessio Bricca, Carsten B. Juhl, Lars Hermann Tang, Søren T. Skou.

**Project administration:** Lars Bo Jørgensen, Søren T. Skou.

**Software:** Carsten B. Juhl.

**Supervision:** Lars Bo Jørgensen, Søren T. Skou.

**Validation:** Lars Bo Jørgensen, Alessio Bricca, Anna Bernhardt, Søren T. Skou.

**Visualization:** Lars Bo Jørgensen, Alessio Bricca, Søren T. Skou.

**Writing – original draft:** Lars Bo Jørgensen, Alessio Bricca, Anna Bernhardt, Carsten B. Juhl, Lars Hermann Tang, Sofie Rath Mortensen, Jonas Ahler Eriksen, Sisse Walløe, Søren T. Skou.

**Writing – review & editing:** Lars Bo Jørgensen, Alessio Bricca, Anna Bernhardt, Carsten B. Juhl, Lars Hermann Tang, Sofie Rath Mortensen, Jonas Ahler Eriksen, Sisse Walløe, Søren T. Skou.

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
