## [Decision Letter · Decision Letter 0]

18 Jul 2022

PONE-D-22-18651Objectively measured physical activity levels and adherence to physical activity guidelines in people with multimorbidity: a systematic review and meta-analysisPLOS ONE

Dear Dr. Jørgensen,

Thank you for submitting your manuscript to PLOS ONE. After careful consideration, we feel that it has merit but does not fully meet PLOS ONE’s publication criteria as it currently stands. Therefore, we invite you to submit a revised version of the manuscript that addresses the points raised during the review process.

ACADEMIC EDITOR:Dear authors,

In addition to the comments made by reviewers, please address the following:

-check the numerical order of the sections of your work (e.g. introduction should be 1 and methods should be 2, and so on);

-section/name "protocol" does exist in PRISMA 2020, please check;

-before synthesis of methods, please add "effect measures";

- after synthesis of methods, please add "Reporting bias assessment"

Thank you

We look forward to receiving your revised manuscript.

Kind regards,

Rafael Franco Soares Oliveira

Academic Editor

PLOS ONE

Journal Requirements:

Additional Editor Comments:

Dear authors,

In addition to the comments made by reviewers, please address the following:

-check the numerical order of the sections of your work (e.g. introduction should be 1 and methods should be 2, and so on);

-section/name "protocol" does exist in PRISMA 2020, please check;

-before synthesis of methods, please add "effect measures";

- after synthesis of methods, please add "Reporting bias assessment".

Thank you

Reviewers' comments:

Reviewer's Responses to Questions

**Comments to the Author**

1. Is the manuscript technically sound, and do the data support the conclusions?

Reviewer #1: Yes

Reviewer #2: Yes

Reviewer #3: Yes

2. Has the statistical analysis been performed appropriately and rigorously? 

Reviewer #1: Yes

Reviewer #2: Yes

Reviewer #3: Yes

3. Have the authors made all data underlying the findings in their manuscript fully available?

Reviewer #1: Yes

Reviewer #2: Yes

Reviewer #3: Yes

4. Is the manuscript presented in an intelligible fashion and written in standard English?

Reviewer #1: Yes

Reviewer #2: Yes

Reviewer #3: Yes

5. Review Comments to the Author

Reviewer #1: First, I want to address my compliments to the authors for the hard work and good job with this paper. It was an exciting topic to explore since the community still uses a lot of indirect quantification. I understand that sometimes it's hard for the laboratories to buy equipment; however, direct methods must be a priority. Moreover, it must be said that the authors were more than cautious about the studies used in this review.

I have only small indications and suggestions.

Lines 157 to 158: it is said that the authors considered medians equivalent to the mean. Could you explain it better, knowing that the variables are different and have different meanings? How could the outcome be different if the variables were the same?

Lines 171 to 172: Instead of "questions not considered relevant," consider replacing it with "questions which were not the same as the aim of the review." If the authors use the first one, readers will never know what questions were not relevant or considered irrelevant by the authors.

Lines 198: It could be better to replace "we performed" for "a narrative synthesis of the results was performed.".

Line 217: Please, replace "was" for "were".

Line 229: Consider the following change - "Fifteen studies were included...".

Lines 284; It could be interesting to add - "% gender and different ethnicities since they are variables known for their massive impact on PA and morbidities.".

Line 295: Typo "conditionswith".

Line 314: Consider replacing "our meta-analysis" with "this meta-analysis.".

Line 323: Same as above.

Lines 335 to 337: May I suggest incentive PA from youth as a preventive measure of multimorbidity instead of promoting while they already are in trouble? It is also necessary at that stage, but it would be better to reinforce the needing to do PA even if people are healthy and not only when they already present symptoms.

Line 344: It could be better to replace - "In the present review".

Line 349: Same as above or similar. It is believed that this review (..).

Lines 353 to 354: Consider replacing for "(...) interpretation, only a few relevant studies were identified (...)".

369: Consider the following - "The present review or similar.".

370: Consider the following - "It was found that".

377: Consider the following - "The review".

Obviously, for this paper, it cannot be done, but maybe consider adding the time of practice as a factor in future reviews.

Remember that pedometers and accelerometers can be tricked (maybe consider it a limitation).

Please, add DOI in the references.

Reviewer #2: Introductio:I believe that the relevance of the study is not properly justified. There is no information that shows whether or not this population practices a recommended amount of physical activity. We assume not, but that information does not exist. There's just this: "Physical inactivity represents an important target in the growing burden of multimorbidity, as it increases the risk of worsening health, development of other chronic conditions (8) and death (9)". Therefore, the objective of the study is not properly substantiated.

Methods: The Kappa index should be used to identify the concordance level between reviewers.

Why ten years as the temporal window for search isn’t fully justified.

Why the option for RCT and observational studies? Why other methodologies aren’t included?

Why the option for those databases?

The subgroup analyses should be justified prior.

Results/Discussion:The results found are not properly justified and analyzed in the discussion. There is only a comparison with the existing bibliography.

Reviewer #3: The article is well constructed and follows the rules for performing a correct systematic review/meta-analysis.

There are only a few points that must be modified:

- Change the Cochrane Handbook reference to just Cochrane (page 5, line 83 and others...);

- In the sample collection process, it is indicated that a comparative pilot study was carried out between reviewers. They must indicate the results of this pilot study;

- In table 2, column 8, sometimes they put a control group/intervention group and other times only control/intervention: they must be standardized;

- The authors begin the discussion of results stating that this is the first systematic review about the level of physical activity in this population - they should not make this statement as they cannot guarantee its veracity...;

- Authors carry out several expressions like "We did", "we carried out", "our review". These terms must be changed throughout the text to remove orality marks.

6. PLOS authors have the option to publish the peer review history of their article (what does this mean?). If published, this will include your full peer review and any attached files.

Reviewer #1: **Yes: **Christophe Domingos

Reviewer #2: No

Reviewer #3: **Yes: **Liliana Ricardo Ramos

---

## [Author Response · Author response to Decision Letter 0]

31 Aug 2022

Dear Editor - Dr. Rafael Franco Soares Oliveira,

We wish to thank both the academic editor and reviewers for their relevant comments on our manuscript; ‘Objectively measured physical activity levels and adherence to physical activity guidelines in people with multimorbidity: a systematic review and meta-analysis’ (PONE-D-22-18651). We have carefully read and provided responses to address all the comments and concerns raised in the peer review and hereby submit a revised version of our paper.

Our detailed point-by-point responses are provided below and all changes are marked in the revised manuscript as requested. We have submitted two versions of the manuscript as requested, one with and another without changes tracked named 'Revised Manuscript with Track Changes' and ‘Manuscript’, respectively - page and line numbers in the response to reviewers are referring to the version of the manuscript with track changes.

Besides increasing the quality and clarity of our paper, we hope that the changes made to the manuscript now makes it suitable for publication in PLOS ONE. We look very much forward to receiving your response to our resubmitted paper.

On behalf of all authors,

Lars Bo Jørgensen, PhD fellow 

Department of Sports Science and Clinical Biomechanics 

University of Southern Denmark

---

## [Decision Letter · Decision Letter 1]

6 Sep 2022

Objectively measured physical activity levels and adherence to physical activity guidelines in people with multimorbidity: a systematic review and meta-analysis

PONE-D-22-18651R1

Dear Dr. Lars Bo Jørgensen,

We’re pleased to inform you that your manuscript has been judged scientifically suitable for publication and will be formally accepted for publication once it meets all outstanding technical requirements.

Kind regards,

Rafael Franco Soares Oliveira

Academic Editor

PLOS ONE

Additional Editor Comments (optional):

Dear authors,

You successfully follow all suggestion made by reviewers and editors. Therefore, our opinion is to accept.

Just please correct the minor details pointed by reviewer 1 in the proof process. 

Congratulations!

Best regards

Reviewers' comments:

Reviewer's Responses to Questions

**Comments to the Author**

1. If the authors have adequately addressed your comments raised in a previous round of review and you feel that this manuscript is now acceptable for publication, you may indicate that here to bypass the “Comments to the Author” section, enter your conflict of interest statement in the “Confidential to Editor” section, and submit your "Accept" recommendation.

Reviewer #1: All comments have been addressed

Reviewer #2: (No Response)

2. Is the manuscript technically sound, and do the data support the conclusions?

Reviewer #1: Yes

Reviewer #2: (No Response)

3. Has the statistical analysis been performed appropriately and rigorously? 

Reviewer #1: Yes

Reviewer #2: (No Response)

4. Have the authors made all data underlying the findings in their manuscript fully available?

Reviewer #1: Yes

Reviewer #2: (No Response)

5. Is the manuscript presented in an intelligible fashion and written in standard English?

Reviewer #1: Yes

Reviewer #2: (No Response)

6. Review Comments to the Author

Reviewer #1: Dear authors,

Thank you for all the modifications. I saw only 2 or 3 inconsistencies in the body text, but overall, it is ready.

Line 44: Fifteen studies (2,172 participants) WAS included -» Fifteen studies (2,172 participants) WERE included;

Lines 141-142: This is really confusing to understand. First, you say "in order to implement any adjustments" and the following sentence states, "no adjustments were made". If the 2 sentences regard each other, don't separate them into 2 sentences. Furthermore, if you're talking about different adjustments, explain what adjustment you're talking about in the second sentence;

Line 272: I would say FIFTEEN rather than FIFTHEEN.

Congratulations,

Reviewer #2: Dear authors,

thank you for addressing my comments, and fully justify your options.

Congratulations on your work.

7. PLOS authors have the option to publish the peer review history of their article (what does this mean?). If published, this will include your full peer review and any attached files.

Reviewer #1: No

Reviewer #2: No

---

## [Editor Report · Acceptance letter]

29 Sep 2022

PONE-D-22-18651R1 

Objectively measured physical activity levels and adherence to physical activity guidelines in people with multimorbidity - a systematic review and meta-analysis 

Dear Dr. Jørgensen:

I'm pleased to inform you that your manuscript has been deemed suitable for publication in PLOS ONE. Congratulations! Your manuscript is now with our production department. 

Kind regards, 

on behalf of

Dr. Rafael Franco Soares Oliveira 

Academic Editor

PLOS ONE